# Evaluation of Histidine-Rich Proteins 2 and 3 Gene Deletions in *Plasmodium falciparum* in Endemic Areas of the Brazilian Amazon

**DOI:** 10.3390/ijerph18010123

**Published:** 2020-12-26

**Authors:** Leandro Góes, Nathália Chamma-Siqueira, José Mário Peres, José Maria Nascimento, Suiane Valle, Ana Ruth Arcanjo, Marcus Lacerda, Liana Blume, Marinete Póvoa, Giselle Viana

**Affiliations:** 1Graduate Program in Epidemiology and Health Surveillance (PPGEVS), Centre for Education and Graduate Programs (NEP), Evandro Chagas Institute (IEC/SVS/MS), 67.030-000 Ananindeua, Pará, Brazil; leandrogoescorrea@gmail.com; 2Parasitology Section, Evandro Chagas Institute-IEC/SVS/MS, 67.030-000 Ananindeua, Pará, Brazil; joseperes@iec.gov.br (J.M.P.); josenascimento@iec.gov.br (J.M.N.); povoamm@gmail.com (M.P.); 3Hemonúcleo Cruzeiro do Sul, State Health Department of Acre, 69.980-000 Cruzeiro do Sul, Acre, Brazil; omsvalle@hotmail.com; 4Central Public Health Laboratory of Amazonas (LACEN/Amazonas), 69.020-245 Manaus, Amazonas, Brazil; anarcanjo@fmt.am.gov.br; 5Heitor Vieira Dourado Tropical Medicine Foundation, 69.040-000 Manaus, Brazil; marcuslacerda.br@gmail.com; 6Leônidas and Maria Deane Institute-Fiocruz Amazônia, 69.027-070 Manaus, Amazonas, Brazil; 7Malaria Technical Group, General Coordination for the Monitoring of Zoonoses and Malaria Vector Transmission Diseases, CGZV Department of Immunization and Communicable Diseases, DEIDT, Health Surveillance Secretariat, SVS, Ministry of Health, 70.070-942 Brasília, Brazil; liana.blume@saude.gov.br; 8National Council for Scientific and Technological Development, CNPq, 71.605-001 Brasília, Brazil

**Keywords:** *Plasmodium falciparum*, rapid diagnostic tests (RDTs), *pfhrp2*, *pfhrp3*, gene deletion

## Abstract

Histidine-rich proteins 2 and 3 gene (*pfhrp2* and *pfhrp3*) deletions affect the efficacy of rapid diagnostic tests (RDTs) based on the histidine-rich protein 2 (HRP2), compromising the correct identification of the *Plasmodium falciparum* species. Therefore, molecular surveillance is necessary for the investigation of the actual prevalence of this phenomenon and the extent of the disappearance of these genes in these areas and other South American countries, thus guiding national malaria control programs on the appropriate use of RDTs. This study aimed to evaluate the *pfhrp2* and *pfhrp3* gene deletion in *P. falciparum* in endemic areas of the Brazilian Amazon. Aliquots of DNA from the biorepository of the Laboratory of Basic Research in Malaria, Evandro Chagas Institute, with a positive diagnosis for *P. falciparum* infection as determined by microscopy and molecular assays, were included. Monoinfection was confirmed by nested-polymerase chain reaction assay, and DNA quality was assessed by amplification of the merozoite surface protein-2 gene (*msp2).* The *pfhrp2* and *pfhrp3* genes were amplified using primers for the region between exons 1 and 2 and for all extension of exon 2. Aliquots of DNA from 192 *P. falciparum* isolates were included in the study, with 68.7% (132/192) from the municipality of Cruzeiro do Sul (Acre) and 31.3% (60/192) from Manaus (Amazonas). Of this total, 82.8% (159/192) of the samples were considered of good quality. In the state of Acre, 71.7% (71/99) showed *pfhrp2* gene deletion and 94.9% (94/99) showed *pfhrp3* gene deletion, while in the state of Amazonas, 100.0% (60/60) of the samples showed *pfhrp2* gene deletion and 98.3% (59/60) showed *pfhrp3* gene deletion. Moreover, 79.8% (127/159) of isolates displayed gene deletion. Our findings confirm the presence of a parasite population with high frequencies of *pfhrp2* and *pfhrp3* gene deletions in the Brazilian Amazon region. This suggests reconsidering the use of HRP2-based RDTs in the Acre and Amazonas states and calls attention to the importance of molecular surveillance and mapping of *pfhrp2/pfhrp3* deletions in this area and in other locations in the Amazon region to guarantee appropriate patient care, control and ultimately contribute to achieving *P. falciparum* malaria elimination.

## 1. Introduction

One of the strategies of the World Health Organization (WHO) to control and subsequently eliminate malaria in countries is timely diagnosis and treatment of the disease [1]. The first-choice diagnosis recommended by the WHO is the use of optical microscopy with the thick blood smear method. However, due to its limitations, such as the need for a trained microscopist to read the slides and electrical energy, it becomes unfeasible for remote areas. Thus, rapid diagnostic tests (RDTs) were developed as an alternative to overcome these limitations by providing rapid and accurate diagnosis [2,3].

Because RDTs are easy to handle and transport, they have become an excellent tool for remote areas and in epidemic situations [1,4]. With a production of 412 million units in 2018, the distribution of RDTs for malaria diagnosis increased annually, expanding the diagnostic capacity in Brazil and contributing to the control and elimination of malaria, according to epidemiological reports [1,2]. Among more than 150 commercially available RDTs, the most sensitive ones for detecting *P. falciparum* malaria, the most severe form of the disease, have the histidine-rich protein 2 (HRP2) as the target molecule. This protein is expressed by both asexual and sexual forms of *P. falciparum* parasite, encoded by the *pfhrp2* gene, which can cross-react with histidine-rich protein 3 (HRP3) due to the structural similarities between them [5,6].

However, studies evaluating the sensitivity and specificity of HRP2-based RDTs showed false-negative results due to the absence of the gene that encodes the HRP2 and HRP3 proteins or for having altered genes encoding histidine-rich proteins (HRPs) 2 and 3, raising questions about the quality and cost-effectiveness of HRP2-based RDTs. In South America, samples collected in Iquitos, Peruvian Amazon, showed > 30.0% *pfhrp2* gene deletion [7]. Similarly, in the analyses of samples collected in the western Brazilian Amazon, a border region of Brazil and Peru, *pfhrp2* and *pfhrp3* gene deletion was observed, but samples from eastern Amazonia did not show the absence of *pfhrp2*. These results were observed concomitantly with an increase in cases of malaria in the Brazilian Amazon region, with 57,221 cases being reported in 2017, representing an increase of 12% over 2016 [8,9].

Given the 37% increase in malaria cases in 2018, which was associated with the increase in records of *P. falciparum* malaria in certain locations in the Brazilian Amazon region compared to the same period of the previous year [8], this study aimed to evaluate *pfhrp2* and *pfhrp3* gene deletions in samples from Brazilian Amazon. This would expand the knowledge on the emergence and spread of gene deletion, providing the malaria control program with much-needed information to target hotspots within states of the Amazon basin.

## 2. Materials and Methods

### 2.1. Samples and Study Design

This retrospective and observational cross-sectional study, without clinical intervention, only considered DNA aliquots of *P. falciparum* previously identified by microscopy (excluding posttreatment slides) from the biorepository of the Malaria Laboratory, Evandro Chagas Institute. These samples were collected from individuals residing in the municipalities of Cruzeiro do Sul in the state of Acre and Manaus in the state of Amazonas between the months of March to October 2016 and 2017. In this period, the autochthonous cases of *P. falciparum* registered were 19,031, and 8496 in Cruzeiro do Sul and Manaus, respectively. Considering this, the goal was to evaluate 150 samples in each municipality (a total of 300 isolates) to determine the frequency of *pfhrp2* and *pfhrp3* gene deletion with a 95% confidence interval (CI) with an error of 5%.

### 2.2. Nested-PCR Reaction for Confirming Plasmodium Species and DNA Quality

DNA aliquots from the biorepository of the Evandro Chagas Institute were processed using the nested-PCR technique for amplifying the minor subunit of the 18S ribosomal RNA (*ssurRNA*) gene with primers previously proposed [10]. Adaptations in PCR conditions were also adopted to confirm the *Plasmodium* species [11]. In order to ensure DNA quality, isolates positive for *P. falciparum* (only monoinfections) except for mixed infections (*P. falciparum* + *P. vivax)* were subjected to amplification of the merozoite surface protein-2 (*msp2* gene), specific for *P. falciparum*, using primers and cycling conditions previously established [12,13,14].

### 2.3. Characterization of pfhrp2 and pfhrp3 Genes

The *pfhrp2* and *pfhrp3* genes were characterized using nested-PCR reactions performed three times by two different technicians (blind testing) to amplify two segments of these genes using primers for the segments extending from the end of exon 1 to the beginning of exon 2 and other primers for the segment including all exon 2 using primers and cycling conditions previously established [6,7]. All PCR analysis was conducted in an automatic thermal cycler. PCR products, positive controls (reference clones 7G8, positive for *pfhrp2* and *pfhrp3* genes; HB3, positive for *pfhrp2* gene and negative for *pfhrp3* gene; and Dd2, negative for the *pfhrp2* gene and positive for the *pfhrp3* gene) and negative controls (sterile distilled water) were subjected to electrophoresis in 2.0% agarose gel (Ultrapure agarose, BRL 155517-014) and were stained with *GelRed* 10,000x in water (GelRed Nucleic Acid Gel Stain, Biotium®, Uniscience LTDA., Fremont, CA, USA). Subsequently, the gel was visualized under ultraviolet light and photographed using a photodocumentation system, producing 250 base pairs (bp) amplicon for *pfhrp2* and 231 bp for *pfhrp3* in the positive samples.

### 2.4. Statistical Analysis

All laboratory results of the *pfhrp2* and *pfhrp3* genes were entered and stored in an electronic database in Microsoft Access, Biostat^®^ 3.5.2 (Microsoft Corporation, Redmond, DC, USA), and were analyzed using *Tableau*^®^ Desktop 2018.3.2 (Tableau Software Inc., Seattle, DC, USA). The chi-squared test (*BioEstat*^®^ version 5.0, Mamirauá Institute, Tefé, Brazil) was used to determine if the parasitemia intervals correlated with the gene deletion profiles observed in the analyzed samples, if there was a difference in *pfhrp2* and *pfhrp3* gene deletion in Acre and Amazonas, and if *pfhrp2* gene deletion was associated with *pfhrp3* gene deletion, in a general way, and in Acre and Amazonas. All analyses used a 95% confidence level, and a *p*-value of ≤0.05 indicated statistical significance.

### 2.5. Ethical Aspects

Only DNA aliquots of *P. falciparum* coming from the research project entitled “*Validação de Método RealAmp* para *o Diagnóstico da Malária em Áreas Endêmicas do Brasil*-Validation of the RealAmp Method for the Diagnosis of Malaria in Endemic Areas of Brazil” were analyzed. The project was approved by the Ethics Committee on Research Involving Humans (CAAE 30488014.4.0000.0019).

## 3. Results

We aimed to include 150 *P. falciparum* isolates from the municipality of Cruzeiro do Sul (Acre) and 150 from Manaus (Amazonas), but only 192 aliquots of DNA of *P. falciparum* attended the inclusion criteria. For the 192 samples, 68.7% (132/192) were from the municipality of Cruzeiro do Sul (Acre), and 31.3% (60/192) were from Manaus (Amazonas). Of this sample, 100.0% of the aliquots of DNA were confirmed with a single infection by *P. falciparum* using the nested-PCR technique for amplifying the 18S rRNA region. DNA quality was ensured by amplifying the *msp2* protein in 75.0% (99/132) of the samples from Cruzeiro do Sul (Acre) and 100.0% (60/60) of the samples from Manaus (Amazonas), making a total of 159 aliquots of DNA subjected to molecular characterization of the *pfhrp2* and *pfhrp3* genes by the amplification of the two segments (the region between exons 1 and 2 and all extension of exon 2).

With regard to the frequency of parasites without HRP2 and HRP3 in the analyzed aliquots of DNA, 71 (71.7%, 95% CI: 60.6–82.8) samples from Cruzeiro do Sul (Acre) were *pfhrp2*-negative, and 60 (100%, 95% CI: 90.6–100.0) samples from Manaus (Amazonas) were *pfhrp2*-negative (Table 1).

*Pfhrp3* deletion was found in 94.9% (94/99, 95% CI: 79.3–100.0) of the aliquots of DNA from Cruzeiro do Sul (Acre) and 98.3% of the aliquots of DNA (59/60, 95% CI: 88.9–100.0) from Manaus (Amazonas). These results are summarized in Table 1.

For all samples analyzed, the frequency of negative parasites for both genes (*pfhrp2* and *pfhrp3*) was 79.8% (127/159, 95% CI: 59.9–99.6). These results can be observed in Table 1, the electrophoretic profile of the *pfhrp2* gene is illustrated in Figure 1, and the distribution of results of *pfhrp2* and *pfhrp3* genotyping by place of origin of the samples is shown in Figure 2 and Figure 3.

## 4. Discussion

The findings of our study confirm that the populations of *P. falciparum* with *pfhrp2* and *pfhrp3* gene deletions are still present in Brazil, with an absolute proportion of negative *pfhrp2* isolates observed in the municipality of Manaus, State of Amazonas, providing the first evidence of possible *pfhrp2* and *pfhrp3* genes deletion in *P. falciparum* isolates from Amazonas state. A percentage of 100.0% of isolates with *pfhrp2*-negative parasites is the highest ever identified in the Amazon countries, comparable only to that observed in populations of *P. falciparum* from Eritrea, a country located in north-eastern Africa, where a 100.0% frequency of *pfhrp2* gene deletion was also observed in the northern region of the Red Sea [15]. It is important to emphasize that these data are preliminary and provide important initial evidence about the deletion for *pfhrp2* and *pfhrp3* in the studied area. However, it is necessary to continue the study, expanding the sampling in order to make it possible to determine the local prevalence and obtain information representative of each municipality as well the Amazon basin [16]. On the other hand, these studies support national malaria programs in strategic planning and operational actions for choosing the RDT that best meets the epidemiological and genetic characteristics of a certain area, since accurate mapping and enhanced monitoring of the prevalence of deletions of *pfhrp2/3* are essential, together with the realization of harmonized methods that allow comparisons between studies [17,18,19].

In the municipality of Cruzeiro do Sul in the state of Acre, a high frequency of negative *pfhrp2* isolates (71.7%) was also observed. This exceeded the percentages found in the Peruvian Amazon, where the prevalence for this event was 41.0%, 1.3 times more than the frequency of 31.2% observed in the same municipality in a prospective study with samples collected in 2012 [5,7]. Although it was not possible to determine the prevalence of negative *pfhrp2* isolates in the municipality of Cruzeiro do Sul, the data from this study confirm the evidence of this event, corroborating previous findings, and reinforce the need for attention in the area, in order to promote expansion and advancement of investigations for accurate and timely actions in public health [2,16,18].

It should be noted that the global frequency of parasites negative for both genes (*pfhrp2* and *pfhrp3*) was also high (79.8%), with a value approximately three times higher than that found in samples from Peru (21–25%) analyzed from 1998 to 2011, demonstrating a rising trend and dispersion of these gene deletions over time. However, to support or refute this hypothesis, further longitudinal studies should be conducted in other locations in Peru, Colombia, and Brazil [2,5,7].

The data of this study suggest that *pfhrp2* and *pfhrp3* gene deletions have a heterogeneous distribution in South America. Furthermore, they show that *pfhrp2*-negative parasites may be concentrated in the contiguous areas of the Amazon basin, which includes the Department of Loreto in the border between the state of Acre and Peru, and the Department of Amazonas in the border between Peru, Brazil, and Colombia [2,7]. Given the above and considering the participation of the Brazilian territory, it can be suggested that the state of Amazonas has a substantial proportion of *pfhrp2*-negative parasites. Aside from the western region of the Amazon countries, it is speculated that the circulating *P. falciparum* parasites in the other regions would have an intact *pfhrp2* gene, which can be identified by HRP2-based RDTs. These considerations confirm the importance of implementing adequate and representative molecular monitoring and surveillance systems to generate reliable data on the prevalence, propagation, origin, and impact of *pfhrp2* and *pfhrp3* gene deletions in a given area, especially in the Amazon basin [20]. It is necessary to highlight that the surveillance of such parasites also depends on the detection of false-negative results by the RDTs assays among suspected malaria cases. This assignment becomes increasingly challenging in the face of pandemic scenarios, like the current COVID-19 pandemic event that impacted malaria in primary care, demobilizing services to attend the demands related to COVID-19, especially in the case of the Amazon basin, an area of low malaria transmission [21].

Although our study did not include samples from the states of Pará, Roraima, and Amapá, the data obtained suggest that *pfhrp2*-negative parasites are confined to the western border regions of Brazil. This hypothesis is supported by the findings of another study carried out in 2017, which did not identify parasites without *pfhrp2* in the state of Pará, in the far east of the Amazon basin, when analyzing the distribution of *pfhrp2* and *pfhrp3* genes deletions in the states of Acre, Pará, and Rondônia in the Brazilian Amazon region [2].

Brazil, Colombia, and Peru seem to constitute a triple border with a geographical location of primordial importance for understanding the aspects involved in the dynamics of the emergence and spread of *pfhrp2* and *pfhrp3* gene deletions in South America. Therefore, investigations aimed at elucidating the contribution of migratory flow (transboundary and internal migration) to the propagation of parasites with gene deletions are crucial [22].

One of the factors that contribute to the increase and propagation of malaria in the Amazon is the migration that occurs in this region, mainly population movements within and outside the Amazon basin. These movements are characteristic of the colonization centers, settlements, and disordered occupation, similar to what occurs in the locations of the Amazon states selected for this study. It is noteworthy that these human populations do not necessarily have a fixed residence, but they move around the interior of the Amazon, aiming at better survival conditions, in a tireless search for job and employment opportunities. Then, the malaria situation in the Amazon region is characterized by significant centers of dispersion and reception of people with *Plasmodium* infection, which perpetuates the transmission in this area and favors the propagation to other Brazilian regions [23,24,25,26].

Thus, it is necessary to continuously (at least every two years) monitor gene deletions throughout the Brazilian territory and better understand the propagation of the migratory flow of people within and outside the Amazon basin to contribute with control and elimination strategies. Currently, the crosscutting approach to knowledge through the inclusion of concepts from paleopathology, to understand the temporal and spatial distribution of networks connecting populations around the world, has been arousing the interest to support control strategies and, mainly, elimination of malaria. These studies generate informative data on the temporal and demographic profiles of malaria occurrence and recommend strategies for infection prevention and treatment of travelers and patients. Furthermore, the data support the implementation of measures for transmission control, health surveillance, and elimination of malaria, according to the epidemiological, ecological, and genetic characteristics of a given area [27,28].

## 5. Conclusions

This study indicated the maintenance of populations of *P. falciparum* with deletion of *pfhrp2* and *pfhrp3* in an area of the Brazilian Amazon, with high percentages (above 30.0%) for this event in the two states of the western region of the Brazilian Amazon basin. In comparison with the percentage of deletions for *pfhrp2-3* observed in other places, the two Brazilian states investigated showed a higher frequency of negative isolates for *pfhrp3*, which reached a value of 98.3% in the municipality of Manaus (Amazonas). In view of these findings, we recommend reconsidering the use of HRP2-based RDTs in the Acre and Amazonas states and emphasize the importance of biennial monitoring the prevalence of these gene deletions in this area and elsewhere in the Amazon Basin. Therefore, further studies are needed to assess the existence, dissemination and impact of these gene deletions, whose presence can lead to delay or loss of treatment, among other public health consequences.

## Figures and Tables

**Figure 1 ijerph-18-00123-f001:**
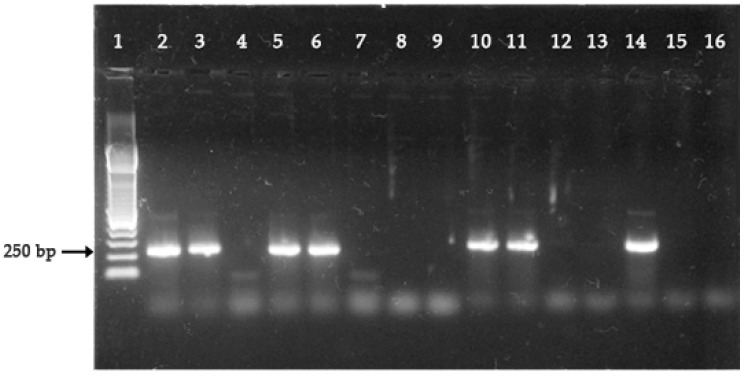
Electrophoretic profile for the molecular analysis of the *pfhrp2* gene (exon 2). Numbers: 1 = 100 bp molecular weight marker; 2, 3, 5, 6, 10 and 11 = positive samples for the *pfhrp2* gene; 4, 7, 8, 9, 12 and 13 = negative samples; 14 = positive pattern HB3; 15 = negative pattern Dd2; 16 = negative control (sterile distilled water).

**Figure 2 ijerph-18-00123-f002:**
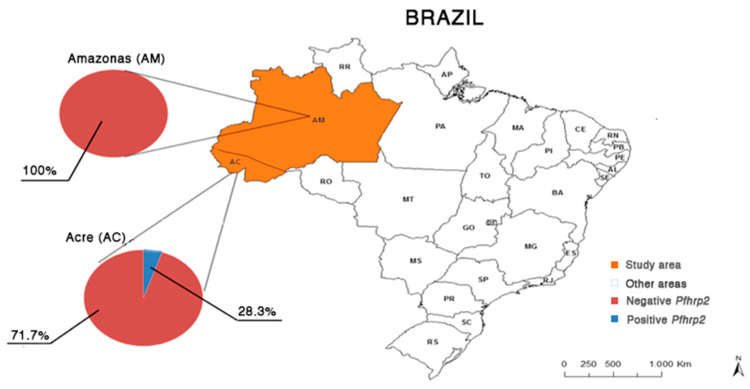
Map of the geographical distribution of the frequency of *pfhrp2*-positive and negative *P. falciparum* isolates observed in the states of Acre (*n* = 99) and Amazonas (*n* = 60), Brazilian Amazon region. Data: continuous cartographic base of Brazil of 2015; Brazilian Institute of Geography and Statistics (IBGE).

**Figure 3 ijerph-18-00123-f003:**
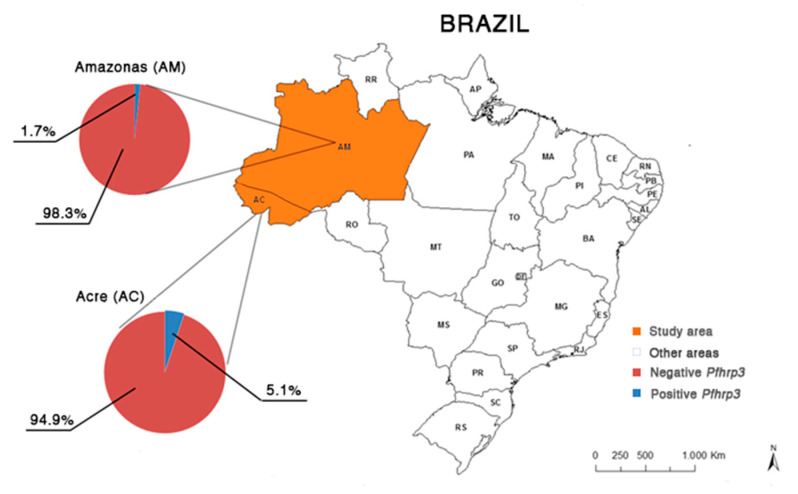
Map of the geographical distribution of the frequency of *pfhrp3*-positive and negative *P. falciparum* isolates observed in the states of Acre (*n* = 99) and Amazonas (*n* = 60), Brazilian Amazon region. Data: continuous cartographic base of Brazil of 2015; Brazilian Institute of Geography and Statistics (IBGE).

**Table 1 ijerph-18-00123-t001:** Summary of genetic markers and distribution of *pfhrp2* and *pfhrp3* frequencies in *P. falciparum* isolates in states of Acre and Amazonas, in Brazil.

States	Acre				Amazonas			
Genetic Markers	*ssurRNA18S*	*msp2*	*pfhrp2* Absence	*pfhrp3* Absence	*ssurRNA18S*	*msp2*	*pfhrp2* Absence	*pfhrp3* Absence
Observed frequencies	100.0% (132/132)	75.0% (99/132)	71.7% (71/99)	94.9% (94/99)	100.0% (60/60)	100.0% (60/60)	100.0% (60/60)	98.3% (59/60)

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
