# Peer review of "Evaluation of Histidine-Rich Proteins 2 and 3 Gene Deletions in Plasmodium falciparum in Endemic Areas of the Brazilian Amazon"

_ijerph, 2020, doi:10.3390/ijerph18010123_

Round 1

Reviewer 1 Report

The manuscript is well written and emphasizes the importance of studies on the monitoring Histidine-rich proteins 2 and 3 genes (pfhrp2 and pfhrp3) deletions that affect the efficacy of RDTs based on HRP2 compromising the correct identification of the P. falciparum species. The possibility of these parasites causes false-negative HRP2-RDT may be left untreated and are at risk of developing complications and/or may further fuel parasite transmission.

The abstract is clearly and accurately describing the content of the article. The methods described are comprehensible. The interpretations and results are consistent. The number of figures and tables is adequate and the data is informative. The discussion is well-founded.

Minor comments: Abstract - Line 22: “Histidine-rich proteins 2 and 3 gene (pfhrp2 and pfhrp3)”, change to pfhrp2 and pfhrp3

Reviewer 2 Report

Dear authors,

I have an opportunity to read your paper on deletion of histidine rich proteins in P. falciparum isolates. My major observations are:

(i) The manuscript requires extensive English editing, especially grammar, etc. 

(ii) Presentation of results could be improved, especially figure 1. 

(iii) Discussion need to be more succint and description of results to be avoided in it. 

All my corrections in the attached word document.

Reviewer 3 Report

The authors assessed pfhrp2-pfhrp3 gene deletion in Brazilian P. falciparum isolates collected in two Amazonian forest regions in 2016-2017. The deletion of these genes has important repercussions for malaria diagnosis since most rapid diagnostic tests (RDT) for P. falciparum malaria are based on HRP2. When both of these genes are deleted, HRP2-based RDT becomes falsely negative, leading to misdiagnosis, untreated malaria infections, and possibly even severe malaria.

The methods are straightforward and are based on PCR protocols developed by other authors. A large majority of parasites had hrp2/3 gene deletion.

More appropriate references should be cited (see my comments below) to support the statements. An addition of few lines on the limitations of the study at the end of the Discussion would be helpful.

The paper is well written and presented.

Major comments:

none

Minor comments:

Line 3: Title of the article : “f” in Falciparum, in small letter

Line 20, 41: pfhrp2 and pfhrp3 in italic

Line 25: “pfhrp2 and” “and” not in italic

Line 30: “msp2” in italic since the authors are referring to the gene

Line 32: Aliquots of DNA from 192 P. falciparum isolates

Line 37: Moreover, 79.8% (127/159) of the isolates displayed gene deletion OR gene deletion was found in 79.8% (127/159) of the isolates

Line 44, keywords: space before “gene”

Line 49, ref 18,19: I suggest that the authors cite WHO documents here. Ref 18 is a PCR diagnostic method.

Line 54, ref: Ref 13 was published 1986, that is before RDT existed. Ref 16 does not seem to support the statement here.

Line 56, ref: Please cite a reference on RDT. Ref 18 was published (in 2002) before WHO started recommending the use of RDT in places where microscopy is not available.

Lines 58-59, ref: The authors are talking about Brazil and its use of RDT. Ref 17 and 18 do not support this statement.

Line 63, ref: Please check if the references are suitable to support the statement.

Line 88: The authors announce here that they planned to analyse 300 isolates, but 192 isolates were actually studied (line 129).

Line 90, 94: Plasmodium, “P” in capital letter

Line 120: “hrp2 and hrp3 gene deletion” and if hrp2 gene…” (hrp in small letters and italicized)

Line 133: Please explain why 25% of samples from Acre were negative for msp2.

Line 140: Delete “about” since an exact number and percentage are given.

Line 142: Add a period at the end of the sentence.

Line 156: pfhrp2 in italic

Line 169: P. falciparum (instead of Plasmodium falciparum)

Line 179: RDT (not TDR for “teste de diagnostic rapido”)

Line 184: “about twice the frequency of 31.2%”: 41% is about 1.3 times more than 31% (not twice).

Line 195: the data… suggest

Line 230: “reception of plasma cell carriers” I am not sure if I understand this part of the statement. Please clarify.
